# Label-Free, Color-Indicating, Polarizer-Free Dye-Doped Liquid Crystal Microfluidic Polydimethylsiloxane Biosensing Chips for Detecting Albumin

**DOI:** 10.3390/polym13162587

**Published:** 2021-08-04

**Authors:** Fu-Lun Chen, Hui-Tzung Luh, Yu-Cheng Hsiao

**Affiliations:** 1Department of Internal Medicine, Division of Infectious Diseases, Taipei Municipal Wan Fang Hospital, Taipei Medical University, Taipei 11031, Taiwan; 96003@w.tmu.edu.tw; 2Department of Internal Medicine, School of Medicine, College of Medicine, Taipei Medical University, Taipei 11031, Taiwan; 3Graduate Institute of Clinical Medicine, National Taiwan University, Taipei 10002, Taiwan; htluh0427@gmail.com; 4Department of Neurosurgery, Shuang Ho Hospital, Taipei Medical University, New Taipei City 23561, Taiwan; 5Taipei Neuroscience Institute, Taipei Medical University, New Taipei City 23561, Taiwan; 6Stanford Byers Center for Biodesign, Stanford University, Stanford, CA 94305-5428, USA; 7Graduate Institute of Biomedical Optomechatronics, College of Biomedical Engineering, Taipei Medical University, Taipei 11031, Taiwan; 8International PhD Program for Biomedical Engineering, Taipei Medical University, Taipei 11031, Taiwan; 9Cell Physiology and Molecular Image Research Center, Wan Fang Hospital, Taipei Medical University, Taipei 11696, Taiwan

**Keywords:** microfluidic, albumin, dye-doped liquid crystal, biosensing chips

## Abstract

We reveal a novel design for dye-doped liquid crystal (DDLC) microfluidic biosensing chips in the polydimethylsiloxane material. With this design chip, the orientation of DDLCs was affected by the interface between the walls of the channels and DDLCs. When the inside of a channel was coated with an N,N-dimethyl-n-octadecyl-3-aminopropyltrimethoxysilyl chloride (DMOAP) alignment layer, the DDLCs oriented homeotropically in a homeotropic (H) state under cross-polarized microscopy. After immobilization of antigens with antibodies on the alignment layer-coated microchannel walls, the optical intensity of the DDLC change from the dark H state to the bright planar (P) state. Using pressure-driven flow, the binding of antigens/antibodies to the DDLCs could be detected in an experimental sequential order. The microfluidic DDLCs were tested by detecting bovine serum albumin (BSA) and its immune-responses of antigens/antibodies. We proved that this immunoassay chip was able to detect BSA antigens/antibodies pairs with the detection limit about 0.5 µg/mL. The novel DDLC chip was shown to be a simple, multi-detection device, and label-free microfluidic chips are presented.

## 1. Introduction

Small-volume, low-cost microfluidic chips have been widely applied due to their rapid detection abilities [1,2]. Unfortunately, the signal of the microfluidic chip is too weak because of the microscale of the biosample. To allow the signals of antigen and antibody response to be more easily detectable, antigen or antibody was labeled with an enzyme [3,4], fluorophore [5,6], or nanoparticle [7,8,9,10,11,12,13,14,15]. However, when the antigen and antibody was binding with the label, the immunobinding response to enhance detectable signals might be affected. Furthermore, the antibody/antigen binding pairs are influenced by being conjugated with the labels [16,17].

Recently, liquid crystal (LC) biosensors have been developed as a new area. Biomolecules cause the LCs to reorient themselves and thus affect their signals. The optical intensity changes from the LCs enable detection by the naked-eye of the label-free biosensors [18]. This reorientation of LCs demonstrates their sensitivity to immunobinding and changes the LC signals [19,20]. In a previous study, LCs as microfluidic devices were also employed to detect the bovine serum albumin (BSA) [21,22]. In addition, the cholesteric LCs (CLCs) have unique optical properties such as Bragg reflection, bistability, and flexibility [23,24,25]. The first CLC biosensor was invented in 2015 [26] in which a high-sensitivity CLC biosensor was shown. However, CLC biosensors require complicated fabrication processes [27,28,29]. To simplify the procedures, a single-substrate device was invented [30]. Furthermore, CLC biosensors can also be integrated with a smartphone, allowing it to detect various diseases at home or in the field [31].

In this paper, we present a dye-doped LC (DDLC)-based microfluidic biosensing chip. The mechanism between antigen/antibody pairs and the DDLCs was investigated. We prove that the DDLC-based multi-microfluidic biosensor differs from a typical biosensor. The antigen/antibody pairs could be detected by measuring the signal intensity of DDLCs in the channel under non-polarized microscopy. The highly sensitive Interface effect between the DDLC molecules and the coated alignment layer composed of DMOAP (N,N-dimethyl-n-octadecyl-3-aminopropyltrimethoxysilyl chloride) was used to detect the BSA antigens/antibodies pairs. The novelty of this paper is that we firstly attempt to design the new DDLC biosensing chip with sensitive, inexpensive, multi-detection, color indicating and non-polarizer properties. A schematic of the design multi-microfluidic DDLC chip is demonstrated in Figure 1.

## 2. Materials and Methods

To generate single-layer cascading microchannels, a 25-μm-thick micro-channel mold was made on a 4-inch (10.2 cm) silicon wafer by using a polydimethylsiloxane (PDMS) soft lithographic fabrication process with a photoresistor. The PDMS was mixed with curing agent and degassed for about 30 min. In addition, the mixture was poured into a master and baked at 65 °C. Next, the PDMS was peeled off from the master and tightly bonded with cleaned substrate by using oxygen plasma treatment. In addition, the nematic LC (E7) mixed with a dichroic dye (PVA black) to form DDLCs was employed in this study. In order to coat the aligned layer of DMOAP, a DMOAP aqueous solution was placed in the microfluidic channels for 30 min, after which the coated channels were washed with deionized water for 1 min. In the immobilization experiment, the BSA solution (0–1 mg/mL) and BSA antibody (0–1000 µg/mL) were immobilized in the alignment layer-coated microchannel. To produce DDLC microfluidic chips, the DDLC material was used to fill empty microfluidic chips at a volume flow rate of 5 µL/min.

## 3. Results and Discussion

### 3.1. BSA Detection Based on the Microfluidic DDLC Chips

The design of the DDLC microfluidic biosensor is shown in Figure 1. We first coated the aligned layer of DMOAP inside the channel as shown in Figure 1a. The BSA and anti-BSA was filled inside the channel, respectively, as demonstrated in Figure 1b,c. Ultimately, the DDLCs were injected into the channel as shown in Figure 1d. The optical image and mechanism of the DDLC microfluidic biosensor is exhibited in Figure 2. Different states could be proposed with and without biomolecules. The vertical alignment layer causes the DDLCs to orient vertically in a homeotropic (H) state to the wall surfaces; the microchannel appeared bright without BSA. When the vertical alignment power was diminished by biomolecules, the H state changed to the planar (P) state, near the channel. The change in intensity could be observed with no polarizer. The DDLC microfluidic biosensor is temperature-independent and can be employed in different situations with a wide temperatures range. To evaluate the relationship between BSA concentrations and images of the DDLC biosensor, BSA was dripped into the DDLC biosensor; the images are demonstrated in Figure 3. The DDLC biosensor is bright with no BSA, and it became darker with increasing BSA concentration. The experimental result shows that the DDLC chips can be used to detect concentrations of BSA. To quantify the data results of the DDLC biosensor, the intensities of the images were analyzed by using software (ImageJ). We used ImageJ software to select the appropriate image range inside the channel and integrate the intensity of pixels to get quantitative value. In Figure 4, the resulting data proved that the intensity of DDLC chips exhibited a linear correlation. Moreover, the DDLC biosensor can successfully be employed to measure BSA concentrations. In addition, volume flow rates of fluids into the microfluidic DDLC chips are important. The different volume flow rates of DDLCs injected into the chip channels have been well studied in the past [32]. A fast volume flow rate (>10 µL/min) resulted in a disordered arrangement of LCs and induced a defective optical texture. In our experiments, we employed a 5 µL/min volume flow rate based on past results [32].

### 3.2. BSA Antibody Immobilized in the DDLC Biosensor Chip

To make the BSA DDLC microfluidic biosensor suitable for clinical use, a BSA antibody (at 0, 10, 100, and 1000 µg/mL) was initially immobilized onto the DDLC microfluidic device. The experimental results show that DDLC microfluidic chips can also be used to test immunocomplexes of BSA/anti-BSA pairs. In addition, the optical intensities of the DDLC microfluidic immunoassay immobilized with 0~10 µg/mL BSA concentrations and 0~1000 µg/mL antibody BSA concentrations were used in the experiment as exhibited in Figure 5. We mixed 0~1000 µg/mL BSA antibody and 0~10 µg/mL BSA antigen to form immune complexes between specific antigen/antibody pairs. Too low of a concentration of the anti-BSA of < 10 µg/mL was unable to induce immunocomplex formation between antigen/antibody pairs. The strength of immune complexes with BSA concentrations of 1 and 10 µg/mL is similar. When the concentration of 100 and 1000 µg/mL of anti-BSA antibodies are mixed, the immunocomplexes resulted in a much-brighter state. Excess concentrations of the anti-BSA changed the orientations of the DDLCs, inducing lower brightness levels. The resulting data show that, compared with the BSA antigen, the BSA immune complex induces a more significant random arrangement of DDLC (Figure 5). The lower concentrations of the anti-BSA for the immunocomplexes could not easily compose the antigen/antibody. However, a higher concentration of the anti-BSA will significantly change the LC arrangement. Therefore, 1 µg/mL of anti-BSA antibody and BSA antigen is the more appropriate concentration. The DDLC biosensor chip can be employed to detect immune complexes and unbound antigens and antibodies. In addition, the linear correlation between the strength of the DDLC chip and the BSA/anti-BSA pairs is shown in Figure 6. We have observed that DDLC has a detection limit of 0.01 µg/mL BSA and 1 µg/mL BSA antibody of immunodetection. These experimental results show that the linear correlation of DDLC-based microfluidic devices can be applied to quantitative immunoassays in the linear range. Compared with well-known immunoassay methods, our microfluidic DDLC chip has color indication, no labeling and is easier to use. Based on the nature of naked-eye detection, this study shows that the DDLC microfluidic biosensor has development potential as a portable biosensing technology for immune detection.

## 4. Conclusions

The DDLC microfluidic biosensing chips are presented in this study. The orientations of DDLCs are affected by the sensitive interface effect between the microchannels and a biomolecule. The DMOAP alignment layer is also coated inside the microchannel. The DDLCs were initially aligned vertically and exhibited a bright H state under the non-polarized microscopy. After the BSA antigens had bound to the BSA antibodies in the microchannel, the optical intensity of the DDLCs transform from bright H to dark P state because of the interruption in the direction of DDLC. Using pressure-driven flow, the BSA antigen/antibody immune complexes can be detected by microscopy. In addition, in the DDLC device, the immunodetection limit of BSA antigen/antibody is 0.01 µg/mL of BSA and 1 µg/mL of anti-BSA. We proved that this microfluidic DDLC immunoassay biosensing chip can detect BSA and antigen/antibody BSA immune complexes through the label-free DDLC immunoassay chip. The new design of this DDLC biosensing chip provides a sensitive, inexpensive, multi-detection, color indicating and non-polarizer system for DDLC-based immunoassays.

## Figures and Tables

**Figure 1 polymers-13-02587-f001:**
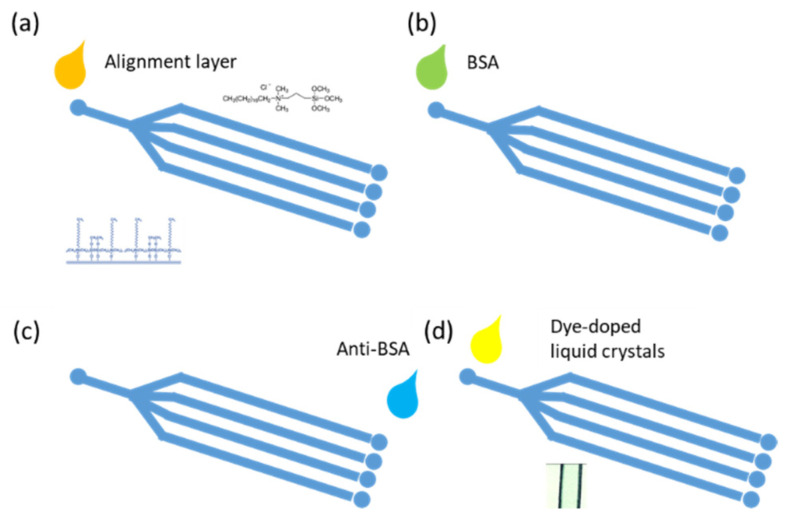
Schematic of microfluidic dye-doped liquid crystal (DDLC) biosensor chips in the presence of bovine serum albumin (BSA) biomolecules.

**Figure 2 polymers-13-02587-f002:**
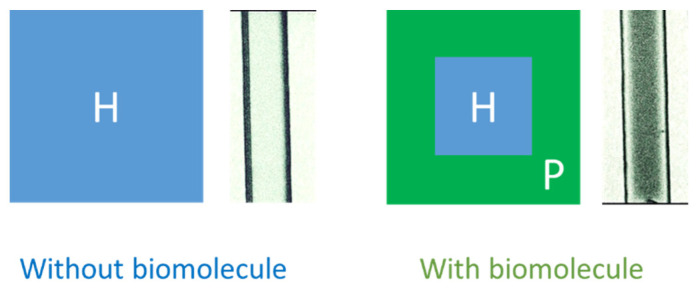
Optical mechanism and images from a non-polarized optical microscope of dye-doped liquid crystal (DDLC) microfluidic chips of a biosensor in both the presence and absence of biomolecules.

**Figure 3 polymers-13-02587-f003:**
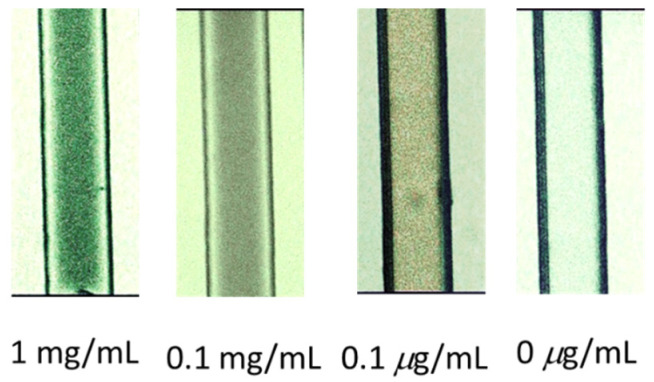
Non-polarized optical images of a dye-doped liquid crystal (DDLC) microfluidic biosensor. The bovine serum albumin (BSA) immobilized at concentrations 0~1 mg/mL.

**Figure 4 polymers-13-02587-f004:**
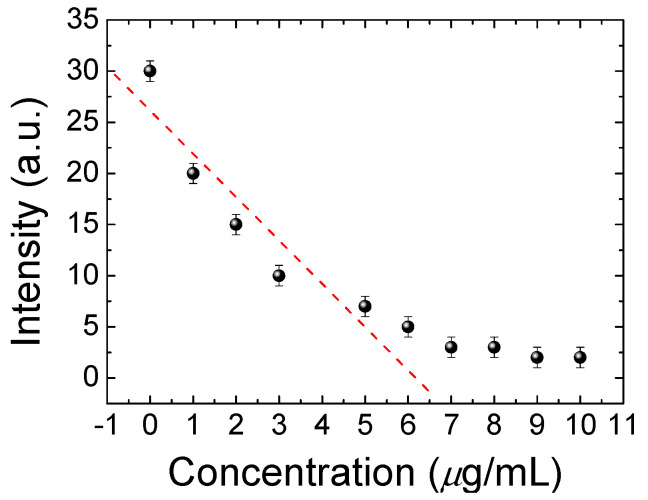
The linear correlations of the intensity of dye-doped liquid crystal (DDLC) multi-microfluidic chips at various concentrations of bovine serum albumin (BSA).

**Figure 5 polymers-13-02587-f005:**
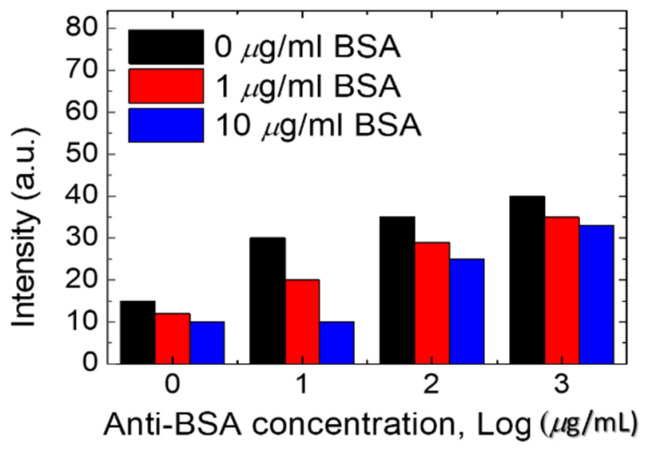
The intensities of immunodetection dye-doped liquid crystal (DDLC) multi-microfluidic chips immobilized with concentrations of 0–10 µg/mL bovine serum albumin (BSA) and concentrations of 0–1000 µg/mL of the BSA antibody.

**Figure 6 polymers-13-02587-f006:**
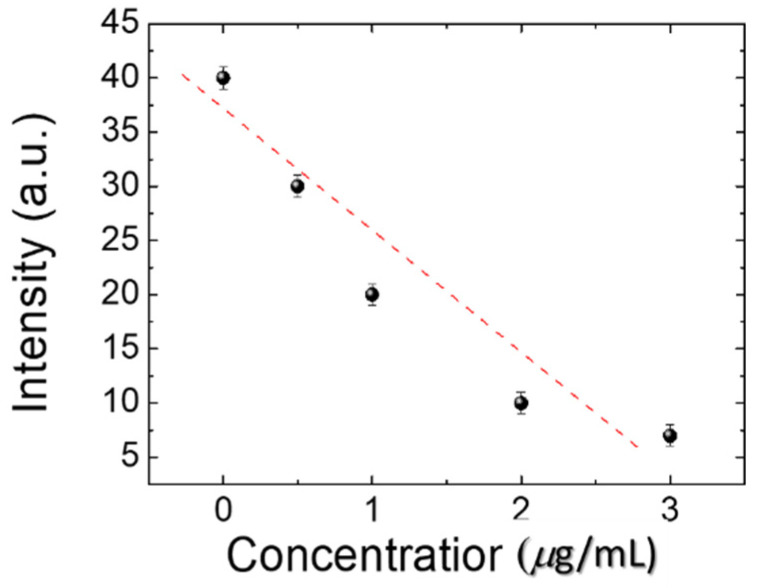
Intensities of dye-doped liquid crystal (DDLC) biosensor chips at various bovine serum albumin (BSA) with a 1 µg/mL of BSA antibody.

## Data Availability

The data presented in this study are available on request from the corresponding author.

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
