# Peer review of "Label-Free, Color-Indicating, Polarizer-Free Dye-Doped Liquid Crystal Microfluidic Polydimethylsiloxane Biosensing Chips for Detecting Albumin"

_polymers, 2021, doi:10.3390/polym13162587_

Round 1
Reviewer 1 Report
Manuscript Number: polymers-1315447-peer-review-v1
Title: Label-Free, Color-Indicating, Polarizer-Free Dye-Doped Liquid Crystal Microfluidic Polydimethylsiloxane Biosensing Chips for Detecting Albumin.
Certainly, the authors have made an important effort to writing the article. There are some comments with this script. Few of them are mention below there for author attention
- Modify the key words to make it more consistent with the manuscript.
- The language of the paper needs revision. There are some grammatical and syntax errors, which must be corrected.
- The curves of figure 4 only contain four data points, which is obviously not enough.
- The figures in this manuscript, like Figure 1, is not clear enough. Please modify it.
- What is the novelty in this work? It should be clearly highlighted.
Author Response
"Please see the attachment."

Reviewer 2 Report
This manuscript presented the design of DDLC microfluidic biosensing chips for BSA detection based on immune-responses of antigens/antibodies. The DDLCs were initially aligned vertically and exhibited a bright H state under the non-polarized microscopy. After the BSA antigens bounding to the BSA antibodies in the microchannel, the optical intensity of the DDLC transform from bright H to dark P state because of the interruption in the direction of DDLC, and provided a color indicating and non-polarizer detection system. After evaluation of the manuscript, I give my comments of minor revision for the manuscript, the following points should be paid attention to:
- The introduction of Fig.1a in section 3.1 is missing.
- Please check the sequence of 1b, c and d. Because the authors demonstrated that “The anti-BSA antibody/BSA mixture was filled in the DDLC biosensor as demonstrated in Fig. 1b and 1c.” The DDLC biosensorwas fabricated firstly.
- The intensities of the images were analyzed by using software(ImageJ)should be more detail in Section 2.
- 5 is confused. It should be described clearly, especially for log of anti-BSAconcentration in text.
Author Response
"Please see the attachment."

Round 2
Reviewer 1 Report
accept